# Automatic Personality Assessment through Movement Analysis

**DOI:** 10.3390/s22103949

**Published:** 2022-05-23

**Authors:** David Delgado-Gómez, Antonio Eduardo Masó-Besga, David Aguado, Victor J. Rubio, Aaron Sujar, Sofia Bayona

**Affiliations:** 1Department of Statistics, Universidad Carlos III, 28903 Getafe, Spain; 2UNED, Universidad Nacional de Educación a Distancia, 28015 Madrid, Spain; aemasob@gmail.com; 3Department of Social Psychology and Methodology, Autónoma University of Madrid, 28049 Madrid, Spain; david.aguado@uam.es; 4Instituto Ingeniería del Conocimiento, Autónoma University of Madrid, 28049 Madrid, Spain; 5Department of Biological and Health Psychology, Autónoma University of Madrid, 28049 Madrid, Spain; victor.rubio@uam.es; 6Department of Computer Engineering, Universidad Rey Juan Carlos, 28933 Madrid, Spain; aaron.sujar@urjc.es (A.S.); sofia.bayona@urjc.es (S.B.); 7Center for Computational Simulation, Universidad Politécnica de Madrid, 28040 Boadilla del Monte, Spain

**Keywords:** personality assessment, movement, Kinect, Big-Five model

## Abstract

Obtaining accurate and objective assessments of an individual’s personality is vital in many areas including education, medicine, sports and management. Currently, most personality assessments are conducted using scales and questionnaires. Unfortunately, it has been observed that both scales and questionnaires present various drawbacks. Their limitations include the lack of veracity in the answers, limitations in the number of times they can be administered, or cultural biases. To solve these problems, several articles have been published in recent years proposing the use of movements that participants make during their evaluation as personality predictors. In this work, a multiple linear regression model was developed to assess the examinee’s personality based on their movements. Movements were captured with the low-cost Microsoft Kinect camera, which facilitates its acceptance and implementation. To evaluate the performance of the proposed system, a pilot study was conducted aimed at assessing the personality traits defined by the Big-Five Personality Model. It was observed that the traits that best fit the model are Extroversion and Conscientiousness. In addition, several patterns that characterize the five personality traits were identified. These results show that it is feasible to assess an individual’s personality through his or her movements and open up pathways for several research.

## 1. Introduction

Personality is a psychological construct aimed at explaining human behavior in terms of a few, stable and measurable individual characteristics [1]. Accurate personality assessments are essential in many areas. For example, in mental health, personality traits have successfully been used to explain individual differences in patients with attention deficit hyperactivity disorder [2,3], addictions [4,5,6], eating disorders [7,8], or suicidal behavior [9,10]. In terms of education, it was found that Conscientiousness, Extroversion, and Neuroticism correlate significantly with exam grades [11]. Other areas where personality analysis is key include organizational [12,13], legal [14,15], and sports [16].

Despite the importance of obtaining accurate personality evaluations, most assessments are conducted using questionnaires and self-reports [17]. Among the advantages of using questionnaires are their ease of use and low cost. In addition, the fact that they are the gold standard in personality assessments means that their psychometric properties have been extensively studied [18,19]. However, various weaknesses have been attributed to scales and questionnaires. One of the greatest shortcomings is faking responses to achieve certain benefits or simply to present a favorable image of oneself [20,21,22]. Another limitation is the number of times a questionnaire can be administered due to learning issues [23]. It has also been indicated that the responses given to the items may be influenced by cultural aspects [24]. These limitations indicate the need to develop new personality assessment strategies that complement the information obtained via questionnaires and self-reports.

In recent decades, several computerized tasks, technologies, and data analysis techniques have been developed to improve personality assessments. For example, the implicit association test, which analyzes reaction times with a computerized classification task, has been used to measure the dimensions of the five-factor model of personality [25]. Voice signals have been recorded and analyzed to estimate personality traits [26]. Text analysis using natural language processing and modern statistical and machine learning techniques has also been proposed [27,28,29]. Social-media analytics is currently a very active field of research [30,31]. Mobile phone logs and wearable technologies are two other areas where personality analyses are being conducted [32,33,34]. Ihsan and Furham recently published an in-depth review of the various modern personality assessment methodologies [35]. All these approaches assume that the different manifestations of the behavior of individuals are expressions of their internal psychological characteristics as the Brunswik’s Lens Model of Perception explains [36].

Another potential predictor of personality is movement [37]. As early as 1933, Allport and Vernon pointed out the relationship between personality and expressive movement [38]. Among the first works that aimed to characterize personality is that developed by Pianesi et al. [39]. In their work, the authors sought to automatically characterize the Extroversion trait of participants who interacted in a meeting while registering speech and a fidgeting measure obtained for the hands, head, and body. Later, Batrinca et al. [40] conducted a related study in which the five dimensions of the Big-Five Personality Model were assessed using backward linear regression with over 29 acoustic and visual non-verbal features obtained from self-presentation short videos (30–120 s). Characteristics were obtained manually with an annotation tool. The visual features corresponded to the length of some actions performed by the participant (e.g., average duration of leaning-back episodes or average duration of frowning). The authors found that Conscientiousness and Extroversion were the best recognizable traits with an Adjusted R^2T^ of 0.188 and 0.172. In a follow-up work, they analyzed personality using short videos (2–5 min) recorded while the participant performed the Map Task [41]. Participants were sitting in front of the computer, and most features were obtained from the head. The reported accuracies ranged from 48% to 81% when predicting the different traits that had been dichotomized into high and low.

However, the lack of affordable mechanisms to characterize movements has hampered research. Fortunately, the improvement of computer vision algorithms for object recognition and the emergence of novel technologies to capture body motion in recent years presents a new scenario for the automatic assessment of personality through movement [42,43]. One of the devices attracting attention is the Microsoft Kinect, which consists of a standard RGB camera, an infrared sensor, an infrared projector, and a set of four microphones. Kinect has achieved an accuracy similar to that of a high-end 3D camera with the benefits of easy portability and low cost [44]. Since its emergence in 2010, several studies have used movement to characterize personality. The aim of this paper is to provide further insight to this field. Therefore, of all the data provided by the Kinect, we will focus on motion.

More recently, Sun et al. estimated the personality of participants while they walked on a red carpet for 2 min [45]. They applied a Fast Fourier Transform to the coordinates captured with a Kinect camera and used stepwise linear regression. The Adjusted R^2^ (and the number of selected variables) were 0.375 (17), 0.375 (10), 0.877 (15) and 0.244 (8) for Extroversion, Psychoticism, Neuroticism, and Social Desirability, respectively. The value of Neuroticism stands out with respect to the results obtained by other authors. The disadvantage of this work is that it is difficult to interpret the selected features because they are Fast Fourier Transform coefficients. Additionally, with Kinect, Furuichi et al. showed that a professional dancer could identify himself/herself based on the skeleton features obtained while performing a short 15-s choreography [46]. They considered the dancers’ performance to be a reflection of personality. Other authors have studied movement with a focus on emotional states rather than characteristics of personality [47]. In many of these works, movement descriptors have been proposed to characterize emotions that could be used as predictors of personality. The most used predictors include movement, speed of movement, and acceleration [48,49,50,51]. A good review on the identification of emotions from movements can be found in the works of Kleinsmith and Bianchi-Berthouze [52] and Witkower and Tracy [53].

Using our approach, we aimed to obtain precise and objective estimates of the individual’s personality in an ecological environment. Similarly to the work of Batrinca et al. [40], we estimated the personality of an individual from the movements made while conducting a short interview. However, these movements were automatically captured by the Kinect camera instead of being manually registered, following the same capture methodology as the works developed by Sun et al. [45] and Furuichi et al. [46]. Another difference with the works of Batrinca et al. [40,41] is that, in our work, all the joints of the body were analyzed instead of only focusing on the upper body [40,41]. In addition, we aimed to evaluate the five traits defined by the Big-Five Personality Model, unlike Pianesi et al. who studied Extroversion alone [39]. Predictors proposed both in the field of personality and in that of emotions were obtained and analyzed using regression with feature selection, similarly to the studies of Batrinca et al. and Sun et al. [40,45].

The following section describes the study set-up, including its design, the characteristics of the sample, and provides a statistical description of the study. We then present the results. The article ends with a discussion of the obtained results and their implications.

## 2. Materials and Methods

Here, we describe the research carried out to predict the personality of an individual from the movements that he/she makes while performing a small interview.

### 2.1. Participants

A group (*N* = 67) composed of undergraduate and master students of psychology from the Universidad Autónoma of Madrid, Spain, (age: Mean = 20.43, SD = 2.62, 77.2% women) participated in the study. Academic credits were awarded for their collaboration, but participants were made aware that the number of credits was fixed and independent of the results. A random identification number was given to each participant to anonymize the data. All participants were informed of the study and signed the required informed consent form. The study was approved by the ethical committee of the Universidad Autónoma of Madrid (CEU-78-1444).

### 2.2. Set-Up

Participants were placed approximately 1.8 m in front of a table on which the Kinect was placed, and behind which was the interviewer. The latter asked the interviewee the following question: “tell me about yourself, which are your hobbies and what are you interested in”. As commented below, the research is similar to that carried out by Batrinca et al. [40]. However, the fact that the participant was standing instead of sitting in front of a computer allowed him/her to move without restriction. This meant that we were able to analyze the movement of all his/her joints. Capturing his/her joints with Kinect allowed us to obtain more accurate measurements of the participant’s joints. The 25 positions that the Kinect camera recorded are displayed in Figure 1. In addition to controlling the location of the participant, the lighting of the room was also taken into account so that 30 frames per second could be obtained instead of the 15 frames that the camera captures when there is poor lighting. Lighting conditions are a key factor that, if inadequate, can produce a lot of noise in the data. In other scenarios with poor illumination or when participants perform greater displacements, it may be convenient to apply strong filtering algorithms such as Kalman filtering [54,55]. The positions of the index fingers and thumbs were not used in this study because they were often not properly located.

### 2.3. NEO Five Factor Inventory

Currently, the Big-Five Personality Model is the dominant paradigm in personality assessments [56]. This is attested to by the fact that 76 of the 81 articles included in the review about personality computing, conducted by Vinciarelli and Mohammadi, adopted this personality model [1].

In this work, the participants filled out the NEO Five Factor Inventory [57]. This scale, made up of 60 5-point Likert-type items, assesses five traits of the participant according to the Big-Five Personality Model. The traits are estimated through 12 items with scores from 0 to 4 points. Each trait is characterized by adding the scores of its related items. Hence, the final score for each trait varies from 0 to 48 points. For example, the following items were included: for the trait of Neuroticism “I often feel tense and nervous”, and for Extroversion “I enjoy partying with lots of people.”

### 2.4. Feature Extraction

According to the existing bibliography in the field of personality and emotions, the raw data were transformed into the different predictors described below. First, the amount of movement was calculated. This predictor was used, among others, by Castellano et al. [48], Batrinca et al. [40], or Sun et al. [45]. In this work, the amount of movement is defined by the following equation:(1)$$QoM_{i}=\frac{\sum_{j}||\vec{p_{i}}\left(t_{j+1}\right)-\vec{p_{i}}\left(t_{j})|\right|}{t}$$
where $\vec{p_{i}}\left(t_{j}\right)$ are the three-dimensional coordinates of the *i*-th joint at time $t_{j}$, ‖ ‖ represents the magnitude of the vector, and $t_{j}$ is the length of the participant’s response. Therefore, 21 predictors were obtained. After that, the linear velocity was calculated. To do so, for each joint *i*, the absolute value of the linear velocity at each instant of time $t_{j}$ was calculated using the following formula:(2)$$vel_{i}\left(t_{j}\right)=\frac{\left|\left|\;\vec{p_{i}}\left(t_{j+1}\right)-\vec{p_{i}}\left(t_{j}\right)\right|\right|}{t_{j+1}-t_{j}}$$

The linear velocity of each joint was characterized through its median and interquartile range (IQR). In this way, 42 predictors were obtained. Similarly, the absolute value of the linear acceleration of each joint *i* was obtained for each instant of time $t_{j}$ [50]. This predictor was defined by:(3)$$acc_{i}\left(t_{j}\right)=\frac{\left||vel_{i}\left(t_{j+1}\right)-vel_{i}\left(t_{j}\right)|\right|}{t_{j+1}-t_{j}}$$

Additionally, similar to the linear velocity, the median and IQR of the above vector magnitudes were calculated. Therefore, 42 predictors were also obtained.

Finally, the angular velocities were calculated for each joint and the different moments of time [46,58]. In particular, the angular velocity was calculated for the following angles:
Elbows: Angles characterized by (Right/Left) Shoulder, (Right/Left) Elbow and (Right/Left) Wrist;Shoulders: Angles characterized by Spine shoulder, (Right/Left) Shoulder, (Right/Left) Elbow;Wrists: Angles characterized by (Right/Left) Elbow, (Right/Left) Wrist and (Right/Left) Hand;Knees: Angles characterized by (Right/Left) Hip, (Right/Left) Knee and (Right/Left) Ankle;Hips: Angles characterized by Spine Base, (Right/Left) Hip and (Right/Left) KneeForward Leaning;Lateral Leaning.


These angular velocities were obtained as follows. First, given a joint *i* and a time instant $t_{j}$, the angular displacement was calculated. Given the three-dimensional coordinates at the instant $t_{j}$ of three joints $\vec{p_{u}}\left(t_{j}\right),\;\vec{p_{i}}\left(t_{j}\right),\vec{\;p_{v}}\left(t_{j}\right)$, the angle formed by the vectors $\vec{u}=\vec{p_{u}}-\vec{p_{i}}$ and $\vec{v}=\vec{p_{v}}-\vec{p_{i}}$ is given by $\theta_{i}\left(t_{j}\right)=acos\left(\frac{\vec{u}\cdot \vec{v}}{\left||\vec{u}||||\vec{v}|\right|}\right)$.

The angular velocity of the *i*-th joint at the instant $t_{j}$ is defined by:(4)$$ang\_vel_{i}\left(t_{j}\right)=\frac{\left|\left|\;\vec{\theta_{i}}\left(t_{j+1}\right)-\vec{\theta_{i}}\left(t_{j}\right)\right|\right|}{t_{j+1}-t_{j}}$$

These angular velocities were summarized by the median and IQR. A total of 24 predictors were obtained corresponding to the median and IQR of the 12 angles previously described. The total number of predictors considered in the study was 130, which were obtained from the previous 129 predictors with the addition of the time that the participant used to answer the question.

### 2.5. Statistical Analysis

Linear regression analysis has widely been used to assess personality using facial images [40], speech [40,59], social media [60,61], and movement [45]. To estimate each of the five traits, stepwise linear regression was used. For each trait, those predictors (from the 130 obtained) that were significant in the simple linear regressions (at the 0.05 level of significance) and satisfied the linearity and homoscedasticity hypotheses were provided to the stepwise linear regression. From these preselected predictors, the stepwise linear regression automatically selected those that achieved the best fit to the trait. In addition, to verify the fit of the model, a chi-square test for normality of the residuals was performed.

## 3. Results

The results obtained are described below. Prior to presenting the most relevant results, and with the aim of facilitating the reader’s understanding, we provide some motion graphs as an example. In this case, Figure 2 shows the movement during the first 30 s of the most distinguishable features for the Extroversion and Consciousness traits for a few participants. More specifically, the first row of the figure shows the head displacement, centered at the origin, of the least extroverted participant and the most extroverted participant (panels A and B, respectively). In addition, the cumulative head displacement as a function of elapsed time is shown (panel C). In these graphs, it can be seen that the most extroverted participant performs a larger head displacement than the least extroverted one. In the second row of the figure, the left knee displacement of the least conscientious participant and the most conscientious participant is shown, along with the cumulative left knee displacement (panels D, E and F, respectively). In this figure, we clearly observed that participants with higher Conscientiousness produce a smaller left knee displacement.

The results obtained for each trait are presented herein. The means (and standard deviations) obtained in the NEO Five Factor Inventory for the different traits are: 18.25 (8.03) for Neuroticism, 32.89 (7.09) for Extroversion, 31.44 (8.13) for Openness to experience, 31.56 (6.71) for Agreeableness, and 32.46 (8.23) Conscientiousness. The average response time of the participants was one minute (61.5 s).

The obtained results for the stepwise linear regression are shown in Table 1. In this table, all the predictors that were selected by the regression for each trait are shown, along with their weights and their *p*-values. In addition, the fit of the model is shown by the R^2^ or Adjusted R^2^ statistic depending on whether one or more variables were selected. The table also shows the *p*-value of the chi-square test for the normality of residuals.

## 4. Discussion

In this work, we analyzed how the automatic registration of movement through the Microsoft Kinect device reflected the way in which individuals express their personality through body movement. It was observed that the traits that were best characterized were Extroversion and Responsibility. These results are in line with those obtained by Batrinca et al. [40]. Nevertheless, in our study, the Adjusted R^2^ values were twice those of the mentioned authors and our study benefits from the fact that the developed system is totally automatic.

The feature that was better characterized by movement is Extroversion (Adjusted R^2^ of 0.39). Specifically, it is related to the movement/displacement of the hands and arms. The movement of the hands was already pointed out as a predictor to characterize Extroversion by Brebner [47]. In addition, it has also been observed that extroversion is characterized by variability in movements of the head. Riggio suggested that extroverts have a higher head movement frequency than introverts and they change their posture more often [62]. Castellano et al. pointed out that emotional expressions in piano players were mainly related to the velocity of the head movements [48]. Pianesi et al. also highlighted head movements as a predictor of personality [39]. The fact that the velocity weight of the right ankle was negative could suggest that extroversion does not correspond to a greater movement of the whole body, rather, it is associated with the upper body.

In the study, we observed that Neuroticism was the trait with the worst adjustment. It obtained an R^2^ of only 7.32%. This result is also in agreement with Batrinca’s work that reported a value of 0.12 [40]. However, these numbers differ from those obtained by Sun et al. who reported an Adjusted R^2^ of 0.877 [45]. This could be because the task is different (talking instead of walking) or simply because our sample does not have high levels of this trait that triggers the patterns. Only the movement of the left wrist is positively related to Neuroticism.

As with Neuroticism, Agreeableness was related only to the movement of the left wrist. However, an interesting fact is that in this case, the relationship was negative. The lesser the movement, the greater the Agreeableness.

The same was also observed for Conscientiousness, which was negatively associated with the movement of the left wrist. Moreover, the fact that the second most discriminating characteristic was the left knee with a negative correlation could indicate that this trait is associated with adopting non-invasive behavior towards the interlocutor, that is, without rapid or abrupt movements. These results agree with those obtained by Jayagopi et al. [63] in which they observed that the total visual activity was a predictor of dominant behavior in group conversations [63].

Finally, individuals who are more Open to the experience tend to provide longer responses (Response Time).

## 5. Conclusions

The results obtained have important theoretical and practical implications. From a theoretical point of view, the findings presented in this study show that it is possible to estimate personality automatically through the way people move, as has been suggested in previous studies [37]. From a practical point of view, as commented in the introduction, having an estimation of the individual’s personality provides benefits in many areas. However, despite the promising results obtained, certain limitations must be pointed out. First, the sample of participants was small and included mostly female students. In future work, it is necessary to repeat this pilot study with a much larger sample and greater representation. Additionally, in our study, we did not register the dominant hemisphere. However, it would be interesting to study if including this factor in the analysis could improve the results. The participants’ movements could also be analyzed while they perform tests that elicit certain personality patterns.

Similar research could also be applied to predict other psychological constructs. It has been observed that it is possible to improve the assessment of an individual’s impulsivity by analyzing the subject’s behavior while performing a continuous performance test based on movement [64,65]. In a related work, Parrado and Ospina used the Kinect with the Iowa gambling test and observed that the time in which the participants made their decisions was significantly related to self-control [66]. It is also possible to analyze participants’ behavior in their natural environment. For example, recently, Sempere-Tortosa et al. observed that the amount of movement of children with ADHD in class is significantly higher than that of their peers without this disorder in almost all joints captured with Kinect [67].

Future work could complement the obtained motion data with a voice analysis, or with a study of the subjects’ emotions recorded with the camera for a richer analysis of personality. Other technologies, such as wearable sensors [68], could also be explored to capture relevant data for these purposes.

Alternatively, the proposed approach could support other fields, such as medicine. The results of this study and the one conducted by Ouyang et al. suggest that this technology could be used to evaluate the effectiveness of treatments [69], and movement analysis could be used to complement other types of diagnostic tests.

The Human Resources Management area is another interesting field of application. Currently, the digitization process to which many organizations are oriented includes the development of automated systems that facilitate decision making. Our results can be used to support the development of automated systems to analyze the self-presentation of candidates in recruitment processes. Likewise, in a context in which work teams are the building blocks on which organizations build their competitive capacity [70], our results can help to understand the interaction between team members. Based on this, a movement analysis could guide training processes to enhance these interactions by improving nonverbal communication.

An interesting application would be to use two Kinect device, one to record the movements of the interviewer/physician and the other to capture the movements of the interviewee/patient, to improve the consultation/selection process via an analysis of the information obtained.

The findings obtained corroborate that it is possible to obtain a good estimation of personality traits based on movement. To conclude, our approach, compared to others present in the literature, has the advantage that it gathers the position of the whole body of the person (not only of the upper part) and that all the variables are registered automatically, leading to an ecological momentary assessment. In addition, the fact that the time cost is low (data acquisition takes approximately 2 min), and the device is affordable (less than $500) facilitates the adoption of the developed system as complementary to information obtained in scales and questionnaires.

## Figures and Tables

**Figure 1 sensors-22-03949-f001:**
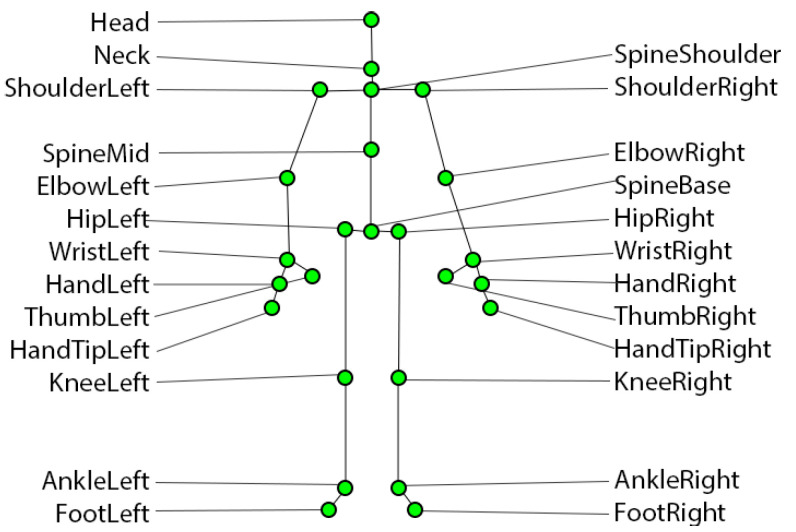
This figure shows the 25 positions captured by the Kinect.

**Figure 2 sensors-22-03949-f002:**
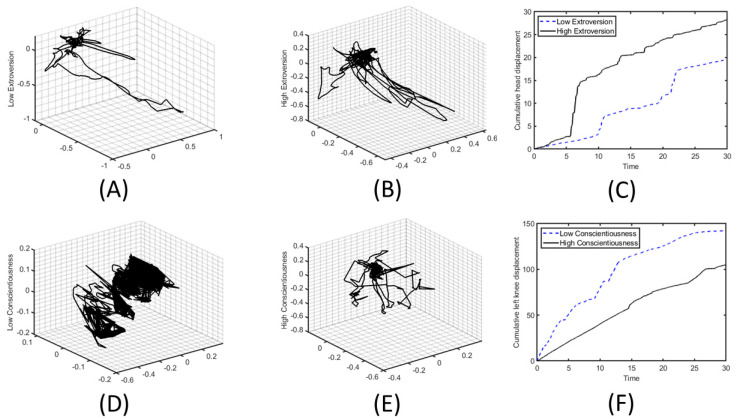
Head and left knee displacement graphs during the first 30 s. (**A**,**B**) Head displacement of the least extroverted and the most extroverted participant, respectively. (**C**) Cumulative head displacement for high and low Extroversion. (**D**,**E**) Left knee displacement of the least and the most conscientious participant, respectively. (**F**) Cumulative left knee displacement for high and low Conscientiousness.

**Table 1 sensors-22-03949-t001:** Coefficients (and *p*-values of the coefficients) obtained using stepwise linear regression for each of the five personality traits. For each trait, the R^2^/Adjusted R^2^ and *p*-value of the chi-square normality test for residuals are shown.

**Neuroticism**
$\hat{N}$ = 18.41 + 1.76 × Median Angular Velocity Left Wrist (*p* = 0.03)
R^2^: 7.32%
chi-square Goodness of fit test for the normality of residuals: 0.04
**Extroversion**
$\hat{E}$ = 35.04 + 31.97 × Median Linear Velocity Right Hand (*p* < 0.01) − 154.91 × Median Linear Velocity Right Ankle (*p* < 0.01) + 159.38 × IQR Linear Velocity Head (*p* = 0.02) − 256.92 × IQR Linear Velocity Right Shoulder (*p* < 0.01) + 24.52 × IQR Linear Velocity Right Ankle (*p* = 0.04) + 0.73 × IQR Linear Acceleration Right Knee (*p* < 0.01) + 1.94 × Median Angular Velocity (*p* = 0.02)
Adjusted R^2^: 39.36%
chi-square Goodness of fit test for the normality of residuals: 0.73
**Openness to experience**
$\hat{O}$ = 26.52 + 2.47 × Median Angular Velocity (*p* = 0.02) + 0.09 × Time Length (*p* = 0.01)
Adjusted R^2^: 12.96%
chi-square Goodness of fit test for the normality of residuals: 0.10
**Agreeableness**
$\hat{A}$ = 31.36 − 2.04 × Median Angular Velocity Left Wrist (*p* < 0.01)
R^2^: 18.93%
chi-square Goodness of fit test for the normality of residuals: 0.89
**Conscientiousness**
$\hat{C}$ = 34.33 − 2.04 × Median Angular Velocity Left Wrist (*p* < 0.01) − 76.44 × Amount of Movement Left Knee (*p* < 0.01) + 1.02 × Median Linear Acceleration Right Ankle (*p* = 0.01) + 0.68 × IQR Linear Acceleration Right Knee (*p* = 0.03)
Adjusted R^2^: 27.88%
chi-square Goodness of fit test for the normality of residuals: 0.34

## Data Availability

Data available on request due to privacy restrictions.

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
