# Peer review of "Automatic Personality Assessment through Movement Analysis"

_sensors, 2022, doi:10.3390/s22103949_

Round 1
Reviewer 1 Report
Your paper is very interesting, clear and well organized. I really appreciated its strong interdiciplinarity
However I must point out that position data obtained by Kinect sensors are typically affected by a strong noise due to criticality in the location of the skeleton joint, change in the enlightement and reflexivity condition, temporary occlusions, etc... So the way you use to compute speed an acceleration of the joints should introduce large values, due to the presence of noise only. Typically in order to apply discrete derivatives to data from Kinect you need a strong filtering algorithm (e.g. Palmieri applied Kalman filters to obtain human arm motion tracking from Kinect measurement, Melchiorre and Scimmi applied filtering algorithms to measure hand velocity from kinetic data in their human-robot hand-over application).
In order to explain the reader how you collected your data you should explain how did you filter row data.
You should also show at least a few examples of time history of the data that you measured, in order to let the unexperienced reader understand which kind of data you are processing
Author Response
"Please see the attachment."

Reviewer 2 Report
-The introduction and literature review seem long and could be shortened.
-The “Related Works” section could be consolidated into the Introduction section before the objectives of this study are stated.
-Sections may need to be renumbered. There are two Sec. 2.
-Some more details on how the experiments were conducted can be included. What’s the average response time or recording time per subject?
-The figure in Appendix A could be moved to the method section.
-There are many data that can be extracted from the Kinect recording. The authors need to justify and explain how they chose predictors.
-More data can be presented in addition to Table 1. What are the numbers of those predictors for other personality traits?
-More discussion on how other readers can benefit from the results of this study can be included.
Author Response
"Please see the attachment."

Reviewer 3 Report
In this manuscript entitled “Automatic personality assessment through movement analysis”, the authors developed a novel system to assess the personality traits of individuals from the movements, which were captured with the low-cost Kinect camera while examinee preforming a small interview. And the research identified the patterns that characterize each of the traits by Big Five model. The research methods were complete and appropriate. Considering this work is of great scientific and future clinical applications, I would like to recommend the publication of this manuscript in Sensors after the following issues are addressed. 1. Full names of all proper nouns should be indicated when the first appear in the text. 2. The section of “Results”, the more details of Table 1 should be added. 3. The size of Table 1 should be revised. 4. The section of More functions such as regression equations listed in data statistics may help the audiences understand this work. 5. Some up to date literatures are recommend to cite: (Engineered Regeneration, 2021, 2, 163-170, https://doi.org/10.1016/j.engreg.2021.10.001.)

Author Response
"Please see the attachment."

Round 2
Reviewer 1 Report
Thank you for your revision, the paper is now ready for publication
Reviewer 2 Report
The authors have addressed all the comments and improved the quality of the manuscript.